

SciPost Phys. Lect.Notes 23 (2021)

# An introduction to kinks in $\varphi^4$-theory

**Mariya Lizunova[1,2] and Jasper van Wezel[2⋆]**

**1** Institute for Theoretical Physics, Utrecht University,
Princetonplein 5, 3584 CC Utrecht, The Netherlands
**2** Institute for Theoretical Physics Amsterdam, University of Amsterdam,
Science Park 904, 1098 XH Amsterdam, The Netherlands

⋆ J.vanwezel@uva.nl

## Abstract

As a low-energy effective model emerging in disparate fields throughout all of physics, the ubiquitous $\varphi^4$-theory is one of the central models of modern theoretical physics. Its topological defects, or kinks, describe stable, particle-like excitations that play a central role in processes ranging from cosmology to particle physics and condensed matter theory. In these lecture notes, we introduce the description of kinks in $\varphi^4$-theory and the various physical processes that govern their dynamics. The notes are aimed at advanced undergraduate students, and emphasis is placed on stimulating qualitative insight into the rich phenomenology encountered in kink dynamics. The appendices contain more detailed derivations, and allow enquiring students to also obtain a quantitative understanding. Topics covered include the topological classification of stable solutions, kink collisions, the formation of bions, resonant scattering of kinks, and kink-impurity interactions.

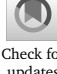
# 1   Introduction

Solitons were introduced into physics by J.S. Russell in 1834 [1], after he observed a solitary wave travelling for miles along the Union Canal near Edinburgh, Scotland, without altering its shape or speed. The dynamics of this particular wave were later described using the Korteweg–de Vries equation [2], but the idea of solitary waves as stable, localized configurations with finite energy in any medium or field [3], turned out to be much more general. They are now known to occur and play an important role in almost all areas of physics, including particle physics [4], cosmology [5,6], (non-linear) optics [7,8], condensed matter theory [9–12], and biophysics [13]. To understand the generic properties of solitary waves, they can be studied in the most elementary models or field theories possible. Two famous examples are the sine-Gordon model and the $\varphi^4$-theory.

   The sine-Gordon model is an integrable model [14], with infinitely many conserved quantities that allow solitary waves in its field configuration, called solitons, to pass through one another while retaining their individual sizes and shapes [15]. It has found many applications including, for example, the analysis of seismic data [16], convecting nematic fluids [17], Josephson-junction arrays [18,19], and magnetic materials [20].

   The $\varphi^4$-theory [21], on the other hand, was first introduced by Ginzburg and Landau as a phenomenological theory of second-order phase transitions [22]. Since then, it has been identified as a low-energy effective description of phenomena in almost any field of physics, making the detailed understanding of its fundamental properties and excitations particularly relevant. The $\varphi^4$-theory can be extended to higher order [23–26], as well as more structured fields [27–32], but the classical scalar theory already contains all essential ingredients required to describe the emergence, dynamics, and interactions of solitary waves called kinks, and will be the focus of these lecture notes.

   The $\varphi^4$-theory is not integrable, and although it possesses stable and localized solitary wave excitations of finite energy [3], these cannot pass through one another unaffected, as they do in the sine-Gordon model [33]. Instead, collisions between kinks may result in a wide array of physical phenomena, such as the excitation of internal modes, resonant and non-resonant scattering [34–37], and the formation of bound states [37,38]. All of these processes have found application throughout physics, in effective descriptions of seemingly disparate things like molecular dynamics [39–41], the motion of domain walls in crystals [42–44], the formation of abnormal nuclei [45–47], and the folding of protein chains [48–50].

   These lecture notes aim to provide a self-contained first introduction to the description of kinks in $\varphi^4$-theory and the rich collection of physical phenomena arising in their dynamics. They are suitable for use as a short course for advanced undergraduate students. Familiarity with basic classical field theory is assumed, and some phenomenological knowledge of, for example, particle physics or basic condensed matter physics will be useful for appreciating the

significance of presented results. Sec. 2 forms the basis of these lecture notes, introducing the classical $\varphi^4$ field theory in $(1+1)$ dimensions, and explaining how non-trivial solutions containing kinks arise. The remaining sections focus on the dynamics and interactions of kinks in the $\varphi^4$-theory, with the phenomenology of kink-antikink collisions being introduced in Sec. 3, followed by the modelling of the resulting scattering and formation of bound states in Secs. 4 and 5, and a discussion of the effect of local disorder in Sec. 6. Exercises appear and the end of most sections, and guide the reader through the main results presented in the text. They are not intended to be challenging. Finally, more detailed discussions of several aspects are presented in appendix A.

We hope these lecture notes provide a basis for understanding some of the ubiquitous phenomena arising throughout effective low-energy descriptions in all realms of physics. They explain why localized excitations in a continuous field behave like massive particles that can scatter, form bound states, and respond to impurities in the continuous medium. They allow you to appreciate the universal nature of these effects, and they prepare you for independently investigating the detailed dynamics of solitary waves in any physical setting.

## 2 Scalar fields in (1+1) dimensions

### 2.1 Topological sectors

Consider a classical, real, and scalar field $\varphi = \varphi(t, x)$ in $(1+1)$-dimensional space-time [33, 51, 52]. Its dynamics is determined by the Lagrangian density:

$$\mathcal{L} = \frac{1}{2}\left(\frac{\partial \varphi}{\partial t}\right)^2 - \frac{1}{2}\left(\frac{\partial \varphi}{\partial x}\right)^2 - U(\varphi). \tag{2.1}$$

The specific potential $U(\varphi) = m^2\varphi^2$ yields a free massive scalar field theory, whose equation of motion is described by the classical Klein-Gordon equation. More generally, the function $U(\varphi)$ can be thought of as a self-interaction potential of the field $\varphi$. We can always use the freedom to choose the zero of energy to ensure that $U(\varphi)$ is a non-negative function of $\varphi$, whose minimum value is precisely zero.

Using the Euler-Lagrange equation, the equation of motion for $\varphi(t, x)$ is found to be:

$$\frac{\partial^2\varphi}{\partial t^2} - \frac{\partial^2\varphi}{\partial x^2} + \frac{dU}{d\varphi} = 0. \tag{2.2}$$

For a static, time independent solution this simplifies to $d^2\varphi/dx^2 = dU/d\varphi$. The dynamics of an initial field configuration $\varphi(t_0, x)$ may be studied by numerically solving the equation of motion on a discrete lattice [45, 53], or employing an appropriate approximation scheme such as the collective coordinate approximation (CCA) [24, 35, 54–57]. Both approaches will be used in the next sections of these lecture notes.

The instantaneous energy of any field configuration $\varphi(t, x)$ is given by the functional:

$$E[\varphi] = \int\limits_{-\infty}^{+\infty}\left[\frac{1}{2}\left(\frac{\partial \varphi}{\partial t}\right)^2 + \frac{1}{2}\left(\frac{\partial \varphi}{\partial x}\right)^2 + U(\varphi)\right]dx. \tag{2.3}$$

Notice that in spite of the time dependence of $\varphi$, the energy $E[\varphi]$ is a time-independent, conserved quantity. The energy of a static ground state field configuration is sometimes referred to as the mass of that field and denoted by $M$. This should not be confused with the Klein-Gordon mass parameter $m$ in a free field theory. For the energy in Eq. (2.3) to be finite, the

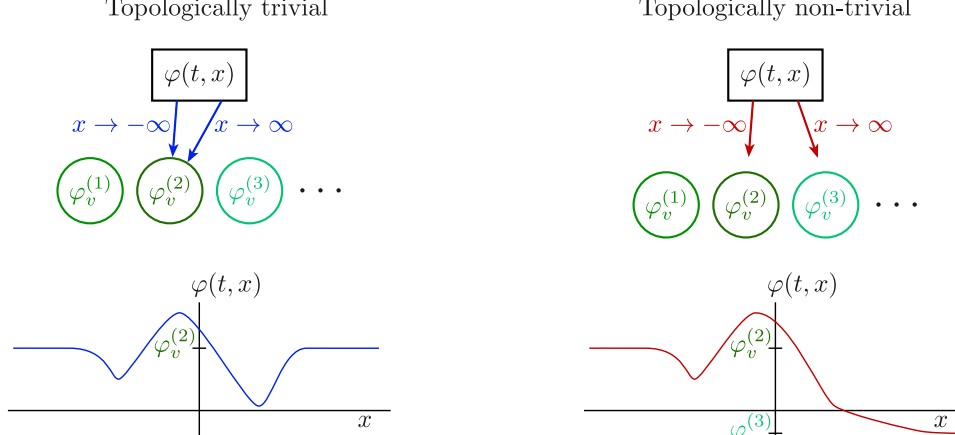

Figure 2.1: Left: Schematic representation and example of the asymptotic behavior for a topologically trivial solution. Right: Schematic representation and example of the asymptotic behavior for a topologically non-trival solution.

integral should converge. This yields the requirement that all physical fields $\varphi(t, x)$ approach a minimum of $U(\varphi)$ sufficiently quickly as $x$ approaches positive or negative infinity.

If there is only a single minimum, at $\varphi = \varphi_v$, then the field configuration $\varphi(t, x) = \varphi_v$ will be the vacuum or ground state of the system. If there are multiple, degenerate minima $\varphi_v^{(1)}$, $\varphi_v^{(2)}$, and so on, they together form a vacuum manifold and any field configuration $\varphi(t, x) = \varphi_v$ is a possible ground state. It is also possible however, to find static field configurations that approach distinct minima at opposing boundaries of space ($x = \pm\infty$). These types of solutions are called topological [51, 52, 58], and more generally, one may divide all static configurations into topological sectors labelled by the set of minima they approach at spatial infinity. This is indicated schematically in Fig. 2.1.

The energy of a static field in any topological sector may be written in a particularly convenient form by introducing the so-called superpotential $W(\varphi)$ [51], defined by:

$$U(\varphi) = \frac{1}{2}\left(\frac{dW(\varphi)}{d\varphi}\right)^2.$$

(2.4)

Notice that we can always find a smooth, continuously differentiable function $W(\varphi)$ satisfying this equation because we assumed $U(\varphi)$ to be a non-negative function of $\varphi$. Using the superpotential, the expression for the energy in Eq. (2.3) can be written for a static field configuration as:

$$
\begin{aligned}
E[\varphi] &= \frac{1}{2}\int_{-\infty}^{+\infty}\left[\left(\frac{d\varphi}{dx}\right)^2 + \left(\frac{dW}{d\varphi}\right)^2\right]dx \\
&= \frac{1}{2}\int\left(\frac{d\varphi}{dx} - \frac{dW}{d\varphi}\right)^2 dx + \int\frac{dW}{d\varphi}\frac{d\varphi}{dx}dx \\
&= \frac{1}{2}\int\left(\frac{d\varphi}{dx} - \frac{dW}{d\varphi}\right)^2 dx + W|_{\varphi(x=+\infty)} - W|_{\varphi(x=-\infty)}.
\end{aligned}
$$

(2.5)

From the final line, it is clear that any field in a given topological sector necessarily has an energy $E \geq E_{\text{BPS}}$, with the minimum possible energy $E_{\text{BPS}} = W|_{\varphi(x=+\infty)} - W|_{\varphi(x=-\infty)}$ named after Bogomolny, Prasad, and Sommerfeld [59, 60]. Any field configuration with energy equal to $E_{\text{BPS}}$ is said to saturate the BPS bound.

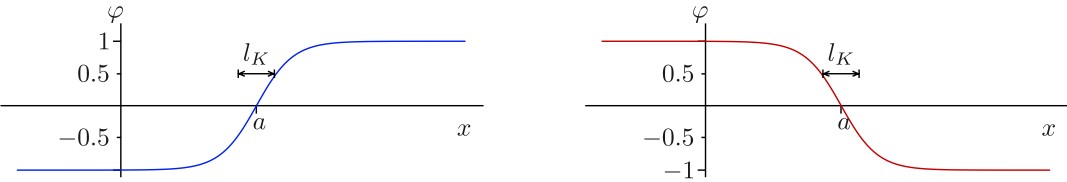

Figure 2.2: The kink (left) and antikink (right) configurations of Eq. (2.9). The characteristic width of the kink, $l_K$, is indicated.

Since the integral in the final line of Eq. (2.5) is over a squared function, the only way to obtain a BPS saturated configuration is to have a field obeying the condition:

$$\frac{d\varphi}{dx} = \frac{dW}{d\varphi} = \sqrt{2U}. \tag{2.6}$$

If such a field configuration does exist, it will be guaranteed by the variational principle to also be a ground state for its topological sector. The Euler-Lagrange equation given by Eq. (2.2) is therefore automatically satisfied by solutions of Eq. (2.6), even though the latter is only a first order differential equation.

## 2.2 Kinks in $\varphi^4$-theory

The simplest scalar field theory having distinct topological sectors is the so-called $\varphi^4$-theory. It is defined as a particular instance of the general model of Eq. (2.1), with the self-interaction potential equal to

$$U(\varphi) = \frac{1}{4}(1 - \varphi^2)^2. \tag{2.7}$$

Here, we write the potential in a dimensionless and mathematically convenient form. The equation of motion with this potential becomes

$$\frac{\partial^2 \varphi}{\partial t^2} - \frac{\partial^2 \varphi}{\partial x^2} + \varphi^3 - \varphi = 0. \tag{2.8}$$

This equation has non-topological or trivial solutions (approaching the same state at both spatial boundaries) given by $\varphi(t,x) = 0$ and $\varphi(t,x) = \pm 1$. These correspond to the field always being at either a maximum or minimum of the potential. The solution $\varphi = 0$, sitting at a maximum, is unstable, while the solutions $\varphi = \pm 1$ form two stable and degenerate ground states.

Non-trivial solutions of Eq. (2.8) lie in a topological sector where the field approaches one minimum as $x \to -\infty$, and the other minimum as $x \to \infty$. One such solution is:

$$\varphi(t,x) = \tanh\left(\frac{x-a}{l_K}\right), \quad \text{with } l_K = \sqrt{2}. \tag{2.9}$$

It is easy to check that this solution satisfies both the equation of motion (2.8), and the BPS condition of Eq. (2.6). It is therefore a stable, non-dissipating configuration, with the minimum possible energy for any field connecting two distinct vacua. This topological solution is often called a kink, and denoted by $\varphi_K$. Another topological solution, called antikink and written $\varphi_{\overline{K}} = -\varphi_K$, connects the same two vacua, but in the opposite direction. As shown in Fig. 2.2, the centre of the kink lies at the (arbitrary) position $x = a$, and it has a characteristic width $l_K$. From Eq. (2.5) the value $M_K = 2\sqrt{2}/3$ can be found for the mass of the kink. Because

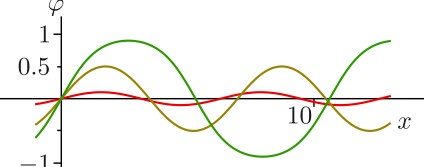
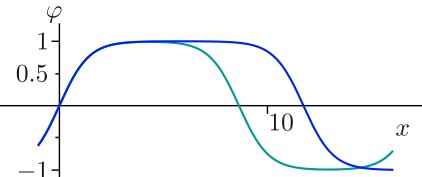

Figure 2.3: The elliptic sine configuration of Eq. (2.11), for different values of the amplitude parameter $\varphi_0$. The left figure contains solutions with (in order of increasing wave length) $\varphi_0 = 0.1$, $\varphi_0 = 0.5$, and $\varphi_0 = 0.9$. The right figure shows the solutions with $\varphi_0 = 0.991$ and $\varphi_0 = 0.999$, in order of increasing wave length.

the energy does not depend on the kink position $a$, translations of the kink in space may be interpreted as zero-energy excitations. There is also a stable excitation mode of the kink at non-zero energy [52, 61], which can be interpreted as an internal or vibrational mode (see appendix A.1). Finally, owing to the Lorentz invariance of Eq. (2.8), the static kink solution may be boosted to yield a dynamical solution in which the kink (or antikink) moves with a constant velocity

$$\varphi(t, x) = \pm \tanh\left(\frac{x - a + vt}{\sqrt{2(1 - v^2)}}\right). \tag{2.10}$$

Here, $v$ is the velocity of the kink measured in units of the speed of light.

You may notice that there is one more static solution to the equation of motion (2.8), given by the elliptic sine:

$$\varphi(t, x) = \varphi_0 \, \text{sn}(bx, k), \quad \text{with } k^2 = \frac{\varphi_0^2}{2 - \varphi_0^2}, \quad b^2 = 1 - \frac{\varphi_0^2}{2}. \tag{2.11}$$

Here, the amplitude $\varphi_0$ is taken to lie between zero and one. In the limit $\varphi_0 \to 1$, the elliptic sine solution approaches the kink configuration (see Fig. 2.3). A more detailed derivation of Eq. (2.11) and its limiting form is given in appendix A.2.

**Exercise 2.1 (Dimensional analysis)** The Lagrangian density defined by Eqs. (2.1) and (2.7) is written in dimensionless form, and can be used to represent the dynamics of many types of physical fields. For example, we can consider a string or piece of rope with displacement waves characterised by the local displacement $u(t,x)$. In this case, the field is a physical quantity with units of length, and the physically relevant Lagrangian density can be written as:

$$\mathcal{L} = \frac{1}{2}\rho\left(\frac{\partial u}{\partial t}\right)^2 - \frac{1}{2}\tau\left(\frac{\partial u}{\partial x}\right)^2 - U(u). \tag{2.12}$$

Here, $\rho$ is the mass density of the string, $\tau$ is the tension, and $U$ can be interpreted as a potential energy density. The potential can be made to have any shape, for example by exposing the string to a gravitational force and placing it on top of a curved surface. Notice that the spatial integral over the Lagrangian density $\mathcal{L}$ is the Lagrangian, with units of energy.

**(a)** Check that all terms in $\mathcal{L}$ have the correct units.

To arrive at a $u^4$-theory, we can place the string on a surface whose height does not change along the length of the string, but which has a double-well shape in the orthogonal direction:

$$U(u) = \rho g h_0 \left(1 - 2\frac{u^2}{r_0^2} + \frac{u^4}{r_0^4}\right). \tag{2.13}$$

Here, the gravitational acceleration is denoted by $g$, the difference in height between the local maxima and minima in the potential is $h_0$, while $\pm r_0$ are the displacement values at which the string reaches a bottom of one of the wells.

**(b)** Introduce dimensionless versions of all physical quantities and show that the Lagrangian density can be written in the form of Eqs. (2.1) and (2.7).

---

**Exercise 2.2 (The kink solution)** Obtain the kink solution of Eq. (2.9) by integrating the BPS equation Eq. (2.6) with the potential given by Eq. (2.7).

---

**Exercise 2.3 (The kink mass)** The value $M_K = 2\sqrt{2}/3$ for the mass of the kink solution $\varphi_K$ can be found using the superpotential $W$.

**(a)** Starting from Eq. (2.7), show that the superpotential can be written as $W = \varphi/\sqrt{2} - \varphi^3/\sqrt{18}$.

**(b)** Since we showed in Exercise 2.2 that the kink is a BPS-saturated solution, we know its energy is given by the BPS form $E_{\text{BPS}} = W|_{\varphi(x=+\infty)} - W|_{\varphi(x=-\infty)}$. Use this to derive the mass $M_K = 2\sqrt{2}/3$.

## 3 Kink-antikink collisions

Field configurations with a kink, like $\varphi_K$ defined by Eq. (2.10), are exact solutions of the equation of motion (2.8) and are therefore stable in the sense that the kink cannot decay or disappear over time. This changes when we consider solutions with multiple kinks [34,45,54].

For example, a field with both a kink moving to the right and an antikink moving to the left is described by:

$$\varphi_{K\overline{K}}(t,x) = \tanh\left(\frac{x+a-v_{in}t}{\sqrt{2(1-v_{in}^2)}}\right) - \tanh\left(\frac{x-a+v_{in}t}{\sqrt{2(1-v_{in}^2)}}\right) - 1. \qquad (3.1)$$

Here, we chose a frame of reference in which the velocities of the two kinks are precisely opposite and equal to $\pm v_{in}$, while the initial positions at $t=0$ equal $\pm a$ and are symmetric around the origin. For brevity, we will from here on omit the explicit distinction between kink and antikink, and refer to the configuration of Eq. (3.1) simply as a field with two kinks. As long as the kinks are far apart, their overlap is negligible and Eq. (3.1) is an exact solution to the equation of motion up to corrections that are exponentially small in $l_K/a$. In other words, the kinks are both stable and both evolve as if they were alone.

When the kinks come close together, however, they start to interact. To see why this must be the case, consider two kinks moving together at high initial speeds, as shown in Fig. 3.1 (how to numerically calculate this time evolution is discussed in appendix A.3). The field configuration starts out with a vacuum solution in most of space, given by $\varphi = -1$ at the edges and $\varphi = 1$ between the kinks. As the kinks move closer together, the middle region shrinks, until the kinks meet at $x = 0$. The kinks may then try and move past one another, but in doing so they create a region of $\varphi = -3$ between the antikink that is now on the left, and the kink on the right. Because $\varphi = -3$ is not a vacuum solution of the $\varphi^4$-theory, the region between the kinks now harbours potential energy. As this region grows, the kinetic energy of the kinks must then decrease, since total energy is conserved. At some point, the kinks halt altogether, reverse their direction of travel, and start moving towards each other again. The region between the kinks now shrinks, and potential energy is converted back into kinetic energy. This time, when the kinks pass through each other, an intermediate region of $\varphi = 1$ is created. Since this does not cost any energy to grow, the kinks can continue and move apart indefinitely. The entire process from beginning to end can be interpreted as a bouncing of two kinks against each other.

## 3.1 Bion formation

The intuitive picture of two kinks behaving like classical particles, with well-defined positions and speeds, whose only effect on each other comes from the order in which they appear in the field, works well as long as the kinks are well-separated. During the times that they overlap, however, the field configuration is no longer close to a solution of the equation of motion, and the two kinks can decay. This can be seen in Fig. 3.1 as the formation of ripples around the edges of the kink, which propagate outwards as time evolves. This decay process can be understood intuitively by considering two kinks with zero velocity that are very close together ($l_K \gg a$). This situation is very close to the non-topological vacuum solution with $\varphi = -1$ everywhere. Because the bump in the field around $x = 0$ is not a solution to the equation of motion (2.8) its energy can dissipate away to infinity, leaving behind a non-topological vacuum, without any kinks. Notice that this same process will also occur if the kinks are initially far apart. However, because the violations of the equation of motion are exponentially small, it will take an exponentially long time for the two-kink configuration to fall apart and relax to the vacuum. Configurations with well-separated kinks are therefore long-lived solutions with lifetimes that can render them effectively stable for all practical purposes.

As long as the initial speeds of two colliding kinks is high, the time in which they overlap is short, and only little energy is dissipated in the form of ripples. Still, the ripples do carry off some energy, and the final speeds at which the kinks move away from one another is lower than $v_{in}$. Considering ever lower initial velocities, there must then be a point at which the

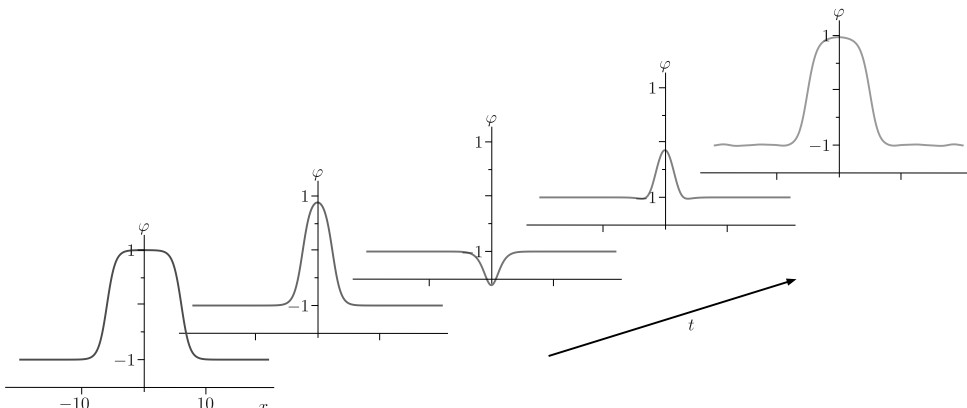

Figure 3.1: The profile of the field $\varphi(t, x)$ for different values of $t$, as a kink and antikink collide. While the kinks attempt to pass through one another, kinetic energy is converted to potential energy in the field and the kinks slow down (three leftmost panels). The kinks then come to a halt (middle panel) and reverse direction, moving away from each other again (two rightmost panels). Some of the initial kineteic energy is radiated away in the form of small ripples (top right panel). For low initial velocities, the outmoving kinks may not have sufficient kinetic energy to escape to infinity, and a bion is formed instead.

colliding kinks can no longer retain sufficient kinetic energy to escape their region of overlap. Below this critical initial velocity, $v_{cr}$, the kinks can still cross, reverse their velocities, cross again, and move apart. However, the kinetic energy is then insufficient to lift the field values between the kinks away from the stable value $\varphi = -1$, and the kinks are halted for a second time. They then again reverse their velocities and move back together. As this process keeps repeating, a localized excitation with large amplitude oscillations is formed [62], as shown in Fig. 3.2 (left panel). This object is often called a bion [33, 63], although it is also sometimes referred to an oscillon [64]. Here, we reserve the term oscillon for a particular low amplitude Gaussian-like solution to the equation of motion that we do not discuss in these lecture notes, and we use the term bion to describe the bound state of kink and antikink discussed here. In (3+1) dimensions, bions are often called quasi-breathers [65]. The bion is a quasi-long-lived state, which will decay to a non-topological vacuum state by emitting ripples or radiation. However, it does so with a very long halftime [66]. We give a more quantitative description of bion formation in the following sections.

## 3.2 Resonances

The phase diagram in Fig. 3.2 shows the outcomes of the kink collisions as a function of their initial velocities. For $v_{in} \geq v_{cr}$, the kinks always bounce and escape to infinity. Having $v_{in} < v_{cr}$, typically results in the formation of a bion. The value of the critical velocity separating these two regions can be established numerically. For an initial separation of $a = 7$, no collisions are observed in the field value at $x = 0$ up to 300 time steps after the initial collision, suggesting the value $v_{cr} \simeq 0.2598$ for the critical velocity [45, 67].

As indicated in Fig. 3.2, ranges of specific values of $v_{in}$ exist that are below the critical velocity, but that nonetheless do not result in the formation of a bion. For such values of the initial velocity, the two kinks start out behaving as if they form a bion, by colliding, separating, reversing velocities, and colliding again. After a fixed number of collisions, however, the two kinks separate completely and escape to infinity, as shown in the top right panel of Fig. 3.2.

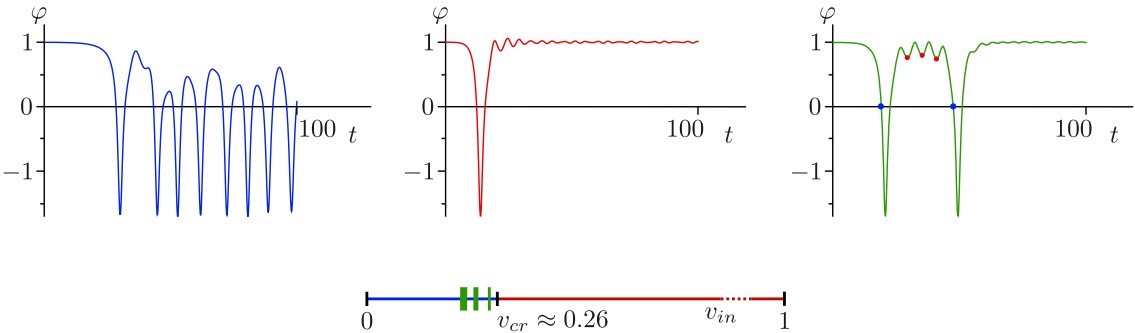

Figure 3.2: The behaviour of the field $\varphi(t,x)$ at $x = 0$ as a function of time for different values of the initial velocity. With $v_{in} = 0.15$ (top left panel) the kink and antikink form a bion and keep colliding, separating, and re-approaching. At $v_{in} = 0.4$ (top center panel) the kinks separate after a single collision, leaving behind small ripples in the field. Finally, for $v_{in} = 0.2384$ (top right panel), kinetic energy is transferred to vibrational modes during the first collision. If the resonance condition of Eq. (3.2) is met, the stored energy can be converted back during a second or later collision, allowing the kinks to escape. The time intervals used in Eq. (3.2) are indicated by blue and red dots. The bottom panel shows a schematic phase diagram of the possible outcomes of kink collisions, as a function of their initial velocities. The left, blue region corresponds to bion formation, the right, red region to single collisions, and the green intervals are reflection windows.

The ranges of $v_{in}$ for which this occurs are called 2-bounce escape windows (or reflection windows), 3-bounce windows, and so on [45].

The escape process is made possible by the presence of an internal, vibrational excitation mode of the kinks [52, 61], whose detailed derivation is discussed in appendix A.1. As two kinks collide, their vibrational modes may be excited, and absorb some of the kinetic energy. If the initial velocity is such that the vibrational motion of the kink profile passes through its equilibrium position precisely as the kinks collide for a second (or third, or later) time, the vibrational energy can be converted back into kinetic energy. This gives the kinks a boost and allows them to escape to infinity. The internal excitations thus effectively act as a storage place for kinetic energy, protected against radiative decay. The resonance condition for efficient conversion of vibrational into kinetic energy is [45]:

$$\omega_R T = 2\pi n + \Delta \quad \Leftrightarrow \quad T = (n + \delta)T_R. \tag{3.2}$$

Here, $\omega_R = 2\pi/T_R$, and $\omega_R$ and $T_R$ are the frequency and the period of the internal kink vibrations (indicated by red dots in the top right panel of Fig. 3.2), and $T$ is the time between the final and penultimate collisions of the kinks (blue dots in the top right panel of Fig. 3.2). The integer $n$ indicates the number of internal oscillations the kink undergoes between collisions, and $\Delta = 2\pi\delta$ is a phase shift incurred during the collision process. Eq. (3.2) thus guarantees the time between collisions is equal to an integer multiple of the period of internal vibrations, up to a constant phase shift. The resonance condition has been numerically checked to both reproduce the analytically derived value of $\omega_R$ (see appendix A.1), and to correctly predict the emergence of escape windows, both in $\varphi^4$ and higher-order theories [26, 45]. In fact, the arrangement of bounce windows for $v_{in} < v_{cr}$ predicted this way, and observed in simulations, is fractal in nature [35].

**Exercise 3.1 (Simulating kink dynamics)** Write a numerical code to calculate the field configuration $\varphi(t, x)$, starting from a configuration at $t = 0$ with only well-separated kinks and no other excitations. The initial state is defined by the positions and velocities of the kinks. Refer to appendix A.3 to set up the code.

**(a)** Starting from a configuration with only a single kink, show that it is stable, and propagates without changing its shape.

**(b)** Consider the initial configuration with two kinks defined in Eq. (3.1), and simulate one case with colliding kinks, and one in which a bion is formed.

**(c)** Identify some resonance windows and use Eq. (3.2) to find $\omega_R$, the frequency of internal kink vibrations.

**(d)** Discuss the limitations of your code, and give a quantitative measure for how well it performs.

*HINT:* As a starting point for finding resonances, sections 2.4, 2.8, and 2.9 in Ref. [33] provide possible sets of parameter values and details about $\omega_R$.

# 4 Collective Coordinate Approximation

Two colliding kinks in $\varphi^4$-theory behave in many aspects like colliding hard-core particles with a short-range attractive interaction. If the kinetic energy of these particles is sufficiently large, the particles will scatter without being affected by their short-ranged attraction. If the kinetic energy is low enough, however, the attraction dominates, and a bound state is formed. This qualitative observation can be made quantitative by constructing a low-energy effective theory for the $\varphi^4$ model, in which only the dynamics of kinks is taken into account, and all other variations in the field are neglected. Such an effective theory is known as the Collective Coordinate Approximation, or CCA [54, 68].

For the sake of concreteness, consider the case of two colliding kinks. In its simplest form, the CCA consists of constructing a theory in which the only degrees of freedom are the positions of the kinks [54]. Going to a frame of reference in which the configuration is symmetric around $x = 0$, there is then only a single degree of freedom $a(t)$ describing the positions of the two kinks. At any point in time, the field configuration associated with a given value of the collective coordinate $a$ is then approximated by:

$$\varphi_{K\overline{K}}(t, x) = \tanh\left(\frac{x + a(t)}{\sqrt{2}}\right) - \tanh\left(\frac{x - a(t)}{\sqrt{2}}\right) - 1. \tag{4.1}$$

The effective Lagrangian for the collective coordinate, $L_{\text{CCA}}(a, \dot{a})$, is obtained by substituting the Ansatz of Eq. (4.1) into the Lagrangian density for the full $\varphi^4$-theory (Eqs. (2.1) and (2.7)) and integrating over space. Taking into account the time dependence of $a(t)$ in the calculation of $\partial\varphi/\partial t$, this leads to a Lagrangian of the form:

$$L_{\text{CCA}}(a, \dot{a}) = \frac{1}{2}m(a)\dot{a}^2 - V(a). \tag{4.2}$$

The position-dependent mass and potential energy in this expression can be written in terms

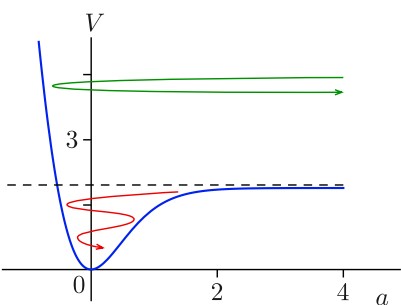

Figure 4.1: The effective potential $V$ in the collective coordinate approximation, as a function of the separation $a$ between kinks. The dashed line at $V = 2M_K$ denotes the asymptotic value of the potential. The green arrow on top schematically indicates an inelastic reflection of the kinks, while the red arrow tending to the bottom of the potential well shows the formation of a quasi-long-lived bion.

of an auxiliary function as:

$$m(a) = I_+(a),$$

$$V(a) = \frac{1}{2}I_-(a) + \frac{1}{4}\int_{-\infty}^{+\infty}(1 - \varphi_{K\bar{K}}^2)^2 dx,$$

$$\text{with} \quad I_\pm(a) = 2M_K \pm \int_{-\infty}^{+\infty} \frac{dx}{\cosh^2\left((x+a)/\sqrt{2}\right)\cosh^2\left((x-a)/\sqrt{2}\right)}. \tag{4.3}$$

At large separation, the integral in the auxiliary functions vanishes, and we find that the mass parameter $m(a)$ and the effective potential $V(a)$ both approach $2M_K$, the mass of two isolated static kinks. For general values of $a$, the integrals can be evaluated numerically, and yield the functional form for the effective potential plotted in Fig. 4.1.

The effective Lagragian of Eq. (4.2) has reduced the initial relativistic field theory to a non-relativistic description of just a single point particle in an external potential. This can be easily solved by considering the Euler-Lagrange equation for the collective coordinate:

$$\frac{d}{dt}\frac{\partial L_{\text{CCA}}}{\partial a} - \frac{\partial L_{\text{CCA}}}{\partial a} = 0$$

$$\Rightarrow \quad m\ddot{a} + \frac{1}{2}\frac{dm}{da}\dot{a}^2 + \frac{dV}{da} = 0. \tag{4.4}$$

The velocity-dependent term in the final line may be interpreted as a frictional force acting on the effective particle. This arises due the change of the effective mass $m(a)$ with position, which is significant only when the two kinks overlap. It is thus a direct manifestation of the two-kink configuration not being a stable solution to the equation of motion (2.8) for the full field theory.

Based on the shape of the effective potential $V(a)$ shown in Fig. 4.1, we can distinguish two qualitatively different types of dynamics. If the initial energy of the particle described by $a(t)$ is sufficiently high, it can cross the potential well around the origin $a = 0$, climb up the potential barrier, which increases linearly with $|a|$ at negative $a$, then turn around and return through the well before escaping to infinity. This describes the bouncing of two kinks. If the initial energy is low enough, however, the particle may lose sufficient energy during its crossing of the potential well to become stuck. It will then oscillate back and forth across $a = 0$ with slowly decreasing amplitude. This describes the formation of a quasi-long-lived bion.

In many cases, numerically solving the equation of motion (4.4) for the effective model yields good agreement with the much more involved numerical solutions for the dynamics

of the full field-theoretical model, both in $\varphi^4$-theory and its higher order generalisations like the $\varphi^6$-model [25]. The Ansatz of Eq. (4.1) does neglect the Lorentz contraction of moving kinks, so that the CCA fails to reproduce relativistic effects that may appear at high velocities. Another limitation of the CCA is that it cannot describe the escape windows in the phase diagram of Fig. 3.2 because the Ansatz does not include a degree of freedom to describe the internal excitation modes of the kinks. Extensions of the CCA that include resonance dynamics are possible [57].

---

**Exercise 4.1 (Simulating effective dynamics)** Write a numerical code that uses the collective coordinate approximation to calculate the field configuration $\varphi(t, x)$, starting from a configuration at $t = 0$ with two well-separated kinks.

  **(a)** Consider the initial configuration with two kinks defined in Eq. (3.1), and simulate one case with colliding kinks, and one in which a bion is formed.

  **(b)** Compare your results to those in part (b) of Exercise 3.1.

  **(c)** Adjust your initial conditions until you start seeing the limitations of the CCA. Quantify the error made in the CCA, as compared to Exercise 3.1.

*HINT:* Some limitations of the CCA are discussed in Section 2.7 of Ref. [33].

---

## 5 Gluing static solutions

The qausi-long-lived bion that we described as a damped oscillator in the previous section, can also be thought of as a combination of parts of three static solutions to the equation of motion. Considering the shape of the field configuration $\varphi(t, x)$ for a bion at some particular time $t$, you may notice that it looks like half a kink and half an antikink, glued together by half a period of the elliptic sine, as shown in Fig. 5.1. In fact, we can make this more precise by constructing a field configuration $\varphi(t, x)$ starting from the static solution in Eq. (2.11). Taking $\lambda$ to be the period of the elliptic sine, we keep only half a period, within the range $-\lambda/4 \leq x \leq \lambda/4$. We then attach half a kink to the left, for $-\infty < x < \lambda/4$, and half an antikink to the right, for $\lambda/4 < x < \infty$. The positions $a = -\lambda/4$ of the kink and $a = \lambda/4$ of the antikink are chosen such that there are no discontinuities in the field. The value of $\varphi_0$, and hence $\lambda$, has to be chosen such that the three static solutions glue together smoothly at the connection points $x = \pm\lambda/4$. We can then consider this configuration of glued static solutions as an Ansatz for the bion field configuration, and calculate its time evolution. This has the advantage of the bion description being independent of the particular process that led to its formations, be it two-kink collisions or otherwise [69]. Notice that in the numerical simulation of the bion dynamics starting from this Ansatz for $\varphi(t, x)$, the field has to be defied both at $t = 0$ and $t = \Delta t$. This can be done by choosing the value of $\varphi_0$ to decrease by a small amount $\delta\varphi_0$ in the second time step. The result of the ensuing time evolution is shown in Fig. 5.1 (right panel), and closely resembles that found in the kink collisions described in Sec. 3. In particular, the dynamics describes a quasi-long-lived state, or bion.

The strategy of gluing together static solutions to find an appropriate Ansatz or initial state may be applied more generally. For example, dynamical kink-kink pairs and triton solutions (kink-antikink-kink) were described this way [70]. The latter may be used as an Ansatz for obtaining the quasi-long-lived solution called a wobbling kink [66, 71].

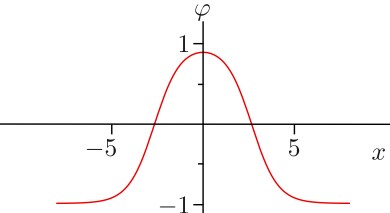
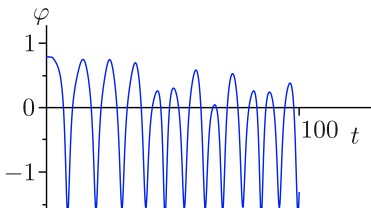

Figure 5.1: The field $\varphi(t,x)$ obtained by gluing together half a kink, half a period of an elliptic sine, and half an antikink. Both the spatial profile at $t = 0$ (left panel), and the its time evolutions (shown at $x = 0$ in the right panel) closely approximate that of the quasi-long-lived bion. For these figures, the parameter values $\varphi_0 = 0.8$ and $\delta\varphi = -0.001$ were used.

## 6 Kink-impurity interactions

The dynamics of kink-antikink collisions arose from taking two exact solutions to the equation of motion in Eq. (2.2), the kink and antikink, and combining them into a single initial state that is not an exact solution. Similarly interesting dynamics may be obtained by taking an exact solution to Eq. (2.2), and using it as in initial state whose dynamics is determined by a slightly different equation of motion. This is the situation encountered in the description of kinks interacting with impurities [37, 72–75].

The $\varphi^4$-theory often arises as a low-energy effective description of some more microscopic model. Imperfections at the microscopic level may then result in local variations of the potential in the effective theory [76–78]. For example, in the context of waves propagating through a solid medium, impurities or defects embedded in the crystal structure may cause impinging waves to scatter or form bound states. To describe such processes, we consider a modification of the potential of Eq. (2.7) by taking:

$$\frac{1}{4}(1-\varphi^2)^2 \longrightarrow \frac{1}{4}(1-\varphi^2)^2(1-\epsilon\delta(x-x_0)). \tag{6.1}$$

Here, $\epsilon$ is the strength of the Dirac delta impurity located at $x = x_0$. For $\epsilon = 0$, the impurity is absent, and the potential equals that of the standard $\varphi^4$-theory. For $\epsilon < 0$, the impurity behaves as a potential barrier, while for $\epsilon > 0$ it acts as a potential well. We may anticipate that a potential barrier will repel kinks, since the energetic price paid for the field value $\varphi$ crossing zero is enhanced at $x = x_0$. The potential well on the other hand lowers the local energy cost of a kink, and will therefore attract it.

To numerically simulate the dynamics of kinks encountering an impurity, we start from the modified equation of motion:

$$\varphi_{tt} - \varphi_{xx} + (\varphi^3 - \varphi)(1-\epsilon\delta(x-x_0)) = 0. \tag{6.2}$$

The Dirac delta function may be approximated on a discrete lattice either by a Kronecker delta with height equal to the inverse of a coordinate step [37, 79], or by a Gaussian profile of the form [67]:

$$\delta(x-x_0) \longrightarrow \frac{1}{\sigma\sqrt{2\pi}}\exp\left[-\frac{1}{2}\left(\frac{x-x_0}{\sigma}\right)^2\right]. \tag{6.3}$$

Here, $\sigma$ is the spatial width over which the impurity affects the potential. It may either be used as a free parameter in the modelling of realistic impurity potentials, or be fixed such that the area under the Gaussian profile equals one [67].

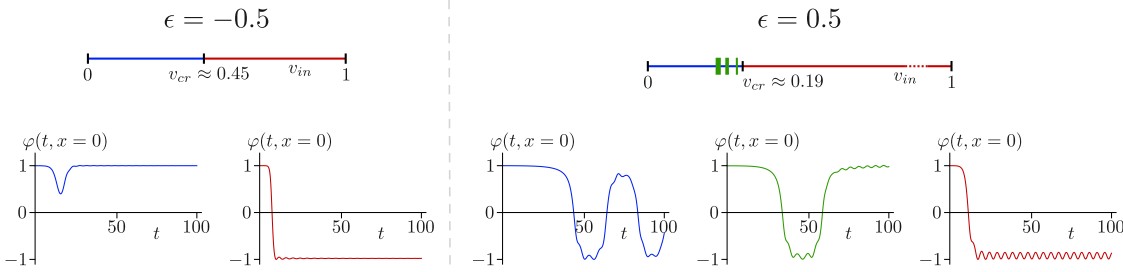

Figure 6.1: Phase diagrams (top row) and example evolutions (bottom row) for kink-impurity collisions with varying initial velocity $v_{in}$ [67]. The left side involves a repulsive impurity of strength $\epsilon = -0.5$, and the right side an attractive impurity with $\epsilon = 0.5$. The example evolutions on the left show the reflection ($v_{in} = 0.4$) and transmission ($v_{in} = 0.8$) of a kink by the repulsive impurity. On the right, they indicate the kink being captured ($v_{in} = 0.1$), being resonantly reflected in a bounce window ($v_{in} = 0.137$), and being transmitted ($v_{in} = 0.5$) by the attractive impurity.

Since the dynamics and physics of kink-impurity interactions closely resemble those of kink-antikink collisions, we give only a brief overview of the possible resulting states. More detailed descriptions can be found in Refs. [37, 67, 72–74, 79]. For the sake of being specific, we consider a single kink described by Eq. (2.10), starting from $a = 6$ and moving towards an impurity located at $x_0 = 0$, with fixed impurity strength $|\epsilon| = 0.5$. We then describe the dynamics for various values of the initial velocity. Fig. 6.1 shows a sketch of the corresponding phase diagram.

For an attractive impurity with $\epsilon = 0.5$, a kink with sufficiently high initial velocity will traverse the impurity and continue propagating on the other side. In crossing the impurity location, it does loose some of its kinetic energy. Some of that energy may be radiated away in the form of small ripples, and some of it is absorbed in an internal, quasi-long-lived excitation mode of the impurity [37, 67].

As the initial velocity of the kink is reduced, at some point a critical velocity $v'_{cr}$ will be encountered, below which the loss of kinetic energy is so large that kinks can no longer traverse the impurity. Typically, the kink is then captured by the impurity, and oscillates around it in a manner similar to the oscillations of kinks and antikinks in a bion. This too is a quasi-long-lived solution to the equations of motion, in the sense that the amplitude of oscillations will slowly decrease to zero. Within a suitably defined CCA, the two types of processes, kinks crossing an impurity and kinks being captured, can again be described as an effective particle with variable mass subject to a resistive force [37].

For specific ranges of initial velocities below the critical value, $v_{in} < v'_{cr}$, the kink may be scattered from an attractive impurity after a finite number of oscillations around it. Analogous to the bounce windows in two-kink collisions, this process may be understood as a resonance between the oscillation time of the kink around the impurity, and the frequency of internal vibration modes [37].

For a repulsive impurity with $\epsilon = -0.5$, kinks of sufficiently high initial velocity are transmitted with some loss of kinetic energy. At low enough initial velocities, kinks again cannot overcome the impurity potential, and are reflected instead. The critical velocity separating the two behaviours, however, does not have the same value as that associated with the attractive impurity, and there are no bounce windows.



---

**Exercise 6.1 (Simulating kink-impurity interactions)** Write a numerical code to calculate the field configuration $\varphi(t, x)$, starting from a configuration at $t = 0$ with a single kink and a single impurity. Refer to appendix A.3 to set up the code.

(a) Consider an initial configuration with the kink at $a = -6$, and a repulsive impurity of strength $\epsilon = -0.5$ at $x_0 = 0$. Simulate its dynamics for several values of the initial velocity. Include at least one velocity for which the kink traverses the impurity, and one for which it is reflected.

(b) Consider an initial configuration with the kink at $a = -6$, and an attractive impurity of strength $\epsilon = 0.5$ at $x_0 = 0$. Simulate its dynamics for several values of the initial velocity. Include at least one velocity for which the kink traverses the impurity, and one for which a bound state is formed.

(c) Identify some resonance windows for $\epsilon > 0$ and use Eq. (3.2) to find the frequencies of the involved internal excitation modes.

*HINT:* As a starting point for finding resonances, section 4 in Ref. [67] and section III in Ref. [37] provide some sets of parameter values and details about frequencies.

---

# 7 Discussion

In these lecture notes, we encountered a wide array of qualitatively different types of dynamics involving kinks in $\varphi^4$-theory. Using straightforward numerical calculations and effective models, we were able to describe effects ranging from the existence of topologically distinct static solutions to the scattering of kinks, and from the formation of quasi-long-lived states to resonant interactions.

This entire range of phenomena arises in the simplest possible classical field theory, which itself is an effective low-energy description for many microscopic models in both solid state and high-energy physics. We hope that advanced undergraduate students studying these lecture notes will come to appreciate both the universality of the rich phenomenology encountered within $\varphi^4$-theory, and the fact that they can qualitatively understand and describe all of it. It may then serve as a first introduction to the powerful and ubiquitous idea of emergence in physics.

**Acknowledgements** This work was performed in the Delta Institute for Theoretical Physics (DITP) consortium, a program of the Netherlands Organization for Scientific Research (NWO), funded by the Dutch Ministry of Education, Culture and Science (OCW).

# A    Appendices

## A.1    Internal excitation mode

To describe internal excitations of any static configuration, we can consider small perturbations on top of the static solution [52]:

$$\varphi(t,x) = \varphi_s(x) + \delta\varphi(t,x). \tag{A.1}$$

Here, $\varphi_s$ is a time-independent field satisfying the equation of motion (2.2). The perturbation $\delta\varphi(t,x)$ may have any shape, but we assume its amplitude $|\delta\varphi| \ll |\varphi_s|$ to be small compared to that of the static field.

Substituting this equation for the field $\varphi(t,x)$ into the equation of motion (2.2), and keeping only terms up to linear order in the perturbation $\delta\varphi(t,x)$, yields the condition:

$$\frac{\partial^2 \delta\varphi}{\partial t^2} - \frac{\partial^2 \delta\varphi}{\partial x^2} - \delta\varphi + 3\varphi_s^2 \delta\varphi = 0. \tag{A.2}$$

This equation can be solved using a separation of variables, by substituting the Ansatz:

$$\delta\varphi(t,x) = \sum_i \psi_i(x)\cos(\omega_i t + \theta_i). \tag{A.3}$$

The temporal phase differences $\theta_i$ between distinct excitation modes will be set to zero from here on without loss of generality. Inserting the Ansatz into the linearised equation of motion results in a set of Schrödinger-like equations, $\hat{H}\delta\psi_i = E_i\delta\psi_i$, with:

$$\hat{H} = -\frac{d^2}{dx^2} + 3\varphi_s^2 - 1,$$
$$E_i = \omega_i^2. \tag{A.4}$$

Thus, the original problem of solving the linearised equation of motion can be restated as the search for all eigenvalues $\omega_i^2$ and eigenstates $\delta\psi_i(x)$ of the linear operator $\hat{H}$. Notice however, that these eigenfunctions accurately describe the excitation modes only in the limit of small amplitude, where ignoring higher order terms in Eq. (A.2) is justified. The exponential dependence of $\delta\varphi$ on $\omega_i$ in Eq. (A.3) implies that any eigenstate of the Hamiltonian with eigenvalue $\omega_i^2 < 0$ corresponds to an instability of $\varphi_s$. If the initial configuration $\varphi_s$ is a stable, static solution to the equation of motion, such negative-energy excitation modes should not occur.

For Lagrangian densities with translational symmetry, there always exists a zero-energy excitation that corresponds to a rigid translation of the initial state [58]. To see this explicitly for the $\varphi^4$-theory, first notice that the Hamiltonian in Eq. (A.4) can be written in terms of the potential given by Eq. (2.7) as:

$$\hat{H} = -\frac{\partial^2}{\partial x^2} + \frac{\partial^2 U}{\partial \varphi^2}. \tag{A.5}$$

Then, compare this to the spatial derivative of the static equation of motion, which can be written as:

$$-\frac{d^2\varphi'}{dx^2} + \frac{d^2 U}{d\varphi^2}\cdot\varphi' = 0. \tag{A.6}$$

Here, $\varphi' = \partial\varphi/\partial x$ is the spatial derivative of the static field $\varphi(x)$. Any solution $\varphi_s$ to the equation of motion, is also a solution to Eq. (A.6). On the other hand, this equation may

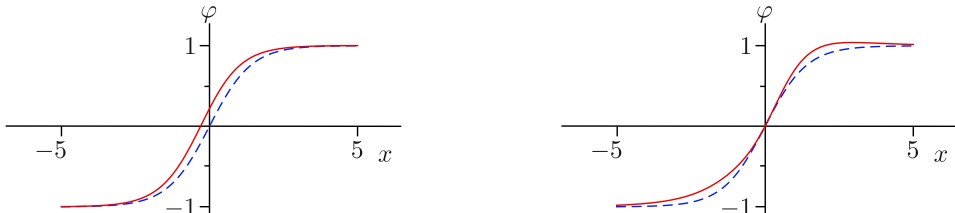

Figure A.1: The spatial profiles of the field $\varphi(t, x)$ for kinks with the internal excitations of Eq. (A.8). The dashed blue lines show a static kink solution. The red line in the left panel adds a zero-energy translational excitation to the kink, while the red line in the right panel shows the excitation of the internal vibrational mode of the kink.

be interpreted as the Schrödinger equation $\hat{H}\varphi' = E\varphi'$ with zero energy and $\hat{H}$ defined by Eq. (A.5). We thus find that for every solution $\varphi_s$ to the equation of motion, there exists a zero-energy excitation $\delta\varphi(t, x) \propto \varphi_s'$.

Focusing now on the specific case of kinks in $\varphi^4$-theory, consider $\varphi_s$ to be the static kink configuration of Eq. (2.9), with $a = 0$. The Hamiltonian whose eigenfunctions are the excitation modes of the kink, then becomes:

$$\hat{H} = -\frac{d^2}{dx^2} - \frac{3}{\cosh^2\left(x/\sqrt{2}\right)} + 2. \tag{A.7}$$

The eigenvalue equation for this specific Hamiltonian happens to be a well-known problem with an analytic solution [61]. There are two eigenfunctions, corresponding to two distinct excitation modes of the kink:

$$\delta\psi_0(x) = \left(\frac{3}{4\sqrt{2}}\right)^{1/2} \frac{1}{\cosh^2\left(x/\sqrt{2}\right)} \qquad \text{with } \omega_0 = 0,$$

$$\delta\psi_1(x) = \left(\frac{3}{2\sqrt{2}}\right)^{1/2} \frac{\sinh^2\left(x/\sqrt{2}\right)}{\cosh^3\left(x/\sqrt{2}\right)} \qquad \text{with } \omega_1 = \sqrt{3/2}. \tag{A.8}$$

The zero-energy solution $\delta\psi_0$ is proportional to $\varphi_s'$, the spatial derivative of the kink solution. We thus recognise it as the translational mode that shifts the kink along the spatial axis. The mode at non-zero excitation energy, $\delta\psi_1$, corresponds to vibrations of the kink around its equilibrium shape, which do not affect the location at which the field value crosses zero. Both excitation modes are plotted in Fig. A.1. The vibrational mode at non-zero energy can be excited during kink collisions, and enables the formation of resonance windows.

> **Exercise A.1 (Kink excitations)** Use Ref. [61] to analytically obtain the eigenvalues and eigenfunctions of the Hamiltonian operator defined in Eq. (A.7).

## A.2 Static solutions

To analytically find static solutions to the equation of motion (2.8) in $\varphi^4$-theory, we assume that there exists a stationary solution $\varphi(x)$ that must then obey the equation of motion with vanishing time derivatives:

$$\frac{d^2\varphi}{dx^2} = \varphi^3 - \varphi. \tag{A.9}$$

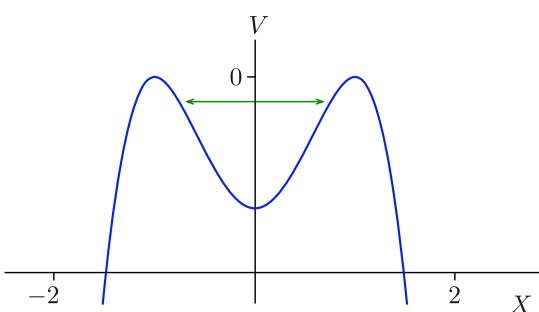

Figure A.2: Sketch of the potential $V(X)$ in Eq. (A.10). The green arrow schematically indicates an oscillating solution. Using the mapping between $X(T)$ and $\varphi(x)$, this corresponds to the static elliptic sine solution in Eq. (A.16) for the $\varphi^4$-theory.

To get some feeling for this equation, notice that it is reminiscent of Newton's second law of motion, $-dV/dX = m\,d^2X/dT^2$, where the role of time $T$ is played by the coordinate $x$, and the position $X$ corresponds the field value $\varphi$. In other words, the configuration $\varphi(x)$ may be read as the trajectory $X(T)$ of a particle with mass $m = 1$, subject to the potential:

$$V(X) = -\frac{1}{4}\left(1-X^2\right)^2.\tag{A.10}$$

This potential is displayed in Fig. A.2. For initial conditions $X(0) = X_0 < X_{\max}$ and $dX/dT = 0$, the particle motion will consist of periodic oscillations between $X_0$ and $-X_0$. These stable trajectories all correspond to static solutions of the $\varphi^4$-theory, after substituting back $X(T) \to \varphi(x)$.

We can find an analytic expression for the static field configurations by first integrating both sides of Eq. (A.9):

$$\frac{d^2\varphi}{dx^2} + \varphi - \varphi^3 = 0,$$
$$\int dx\,\frac{d\varphi}{dx}\left(\frac{d^2\varphi}{dx^2} + \varphi - \varphi^3\right) = \int dx\,\frac{d\varphi}{dx}(0),$$
$$\frac{1}{2}\left(\frac{d\varphi}{dx}\right)^2 + \frac{1}{2}\varphi^2 - \frac{1}{4}\varphi^4 = C.\tag{A.11}$$

Here, $C$ is a constant of integration. Using the intuition based on the particle trajectory analogy, we can choose boundary conditions such that $\varphi = \varphi_0$ at the point where $d\varphi/dx = 0$. This yields $C = \varphi_0^2/2 - \varphi_0^4/4$, and suggests separating the amplitude and spatial dependence of the field, by writing $\varphi(x) = \varphi_0 \chi(x)$, with $|\chi(x)| \leq 1$. In terms of these new variables, Eq. (A.11) becomes:

$$\left(\frac{d\chi}{dx}\right)^2 = 1 - \chi^2 - \frac{1}{2}\varphi_0^2\left(1-\chi^4\right)$$
$$\equiv b^2(1-\chi^2)\left(1-k^2\chi^2\right).\tag{A.12}$$

In the final line, we introduced the definitions:

$$k^2 = \frac{\varphi_0^2}{2-\varphi_0^2},$$
$$b^2 = 1 - \frac{\varphi_0^2}{2}.\tag{A.13}$$

Because we know from the particle trajectory analogy that there will be no stable solutions for $|\varphi_0| > 1$, we always have $0 \leq k^2 \leq 1$, and $1/2 \leq b^2 \leq 1$. Integrating Eq. (A.12) now yields:

$$\left(\frac{d\chi}{dx}\right)^2 = b^2(1-\chi^2)\left(1-k^2\chi^2\right),$$

$$\int_0^{x'} dx \, \frac{|d\chi/dx|}{\sqrt{(1-\chi^2)(1-k^2\chi^2)}} = \int_0^{x'} dx \, b,$$

$$\int_{\chi(0)}^{\chi(x')} \frac{d\chi}{\sqrt{(1-\chi^2)(1-k^2\chi^2)}} = bx'. \tag{A.14}$$

In the final line, we assumed that $d\chi/dx$ is positive over the integration interval. It is easily checked that the final solution we obtain below will be valid also along intervals where $d\chi/dx$ is negative, and that these intervals connect smoothly. Choosing $\chi(0) = 0$ without loss of generality, and making the substitutions $\chi \equiv \sin(\psi)$ and $C' \equiv \arcsin \chi(x')$, Eq. (A.14) reduces to:

$$\int_0^{C'} \frac{d\psi}{\sqrt{\left(1-k^2\sin^2\psi\right)}} = bx'. \tag{A.15}$$

The integral on the left hand side is a standard integral, known as the incomplete elliptic integral of the first, and is often denoted by $F(C', k)$ [80, 81]. Using the known properties of the elliptic integral, Eq. (A.15) can be inverted, and yields the solution $\chi(x') = \text{sn}(bx', k)$, with $\text{sn}(bx, k)$ the elliptic sine [80, 81]. The static solutions to the equation of motion for $\varphi^4$-theory can thus finally be written as:

$$\varphi_s(x) = \varphi_0 \, \text{sn}(bx, k). \tag{A.16}$$

For $\varphi_0 < 1$, these solutions are the elliptic sines shown in Fig. 2.3 of the main text. For very small values of $\varphi_0$, the elliptic sine approaches $\sin(x)$, and the static solution corresponds to small sinusoidal oscillations around the unstable homogeneous solution $\varphi = 0$, which it approaches in the limit $\varphi_0 \to 0$.

To understand the opposite limit, of $\varphi_0 \to 1$, notice that the period of the static solution $\varphi_s(x)$ is given by [81]:

$$T = \frac{4\,F(\pi/2, k)}{\sqrt{1-0.5\varphi_0^2}}. \tag{A.17}$$

The period is plotted as a function of $\varphi_0$ in Fig. A.3. As $\varphi_0$ approaches one, the period diverges. In terms of the particle trajectory analogue, this situation corresponds to the massive particle starting out precisely in an unstable equilibrium state on one of the potential maxima. If the particle at some point (spontaneously, or due to an infinitesimal perturbation) leaves the unstable maximum, it will traverse the minimum in the potential and ends up at the opposite maximum at infinite time.

In terms of the field $\varphi_s(x)$, the same limiting behaviour corresponds precisely to a kink configuration. Taking the limit $\varphi_0 \to 0$ directly in Eq. (A.14), and using $k^2 \to 1$ and $b \to 1/\sqrt{2}$, we see

$$\int_0^{\chi(x')} \frac{d\chi}{1-\chi^2} = \frac{x'}{\sqrt{2}},$$

$$\left[\frac{1}{2}\ln\left(\frac{1+\chi}{1-\chi}\right)\right]_0^{\chi(x')} = \frac{x'}{\sqrt{2}},$$

$$\chi(x') = \tanh\left(x'/\sqrt{2}\right). \tag{A.18}$$

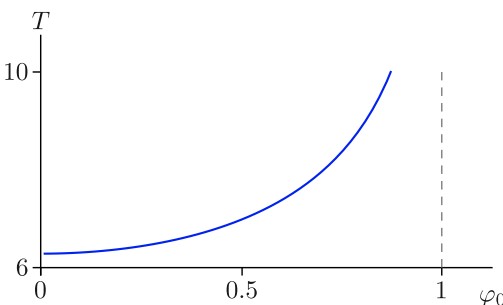

Figure A.3: The dependence of the elliptic sine period $T$ in Eq. (A.17) on the value of the amplitude parameter $\varphi_0$. The period tends to infinity as the amplitude approaches one.

Since we took $\varphi_0 \to 1$, the final static solution $\varphi_s$ equals $\chi$, and we recover the equation for a kink, Eq. (2.9), centered at the origin.

---

**Exercise A.2 (Static solutions)** Reproduce the derivations of the elliptic sine and kink solutions in this appendix.

---

## A.3 Numerical method

We would like to numerically integrate the equation of motion (2.2) and determine the time evolution of any given initial field configuration. To do so, we are first of all forced by the finite computing power of any numerical code to restrict our attention to a finite interval of space and time. We thus consider only positions $-x_{\max} \leq x \leq x_{\max}$, and times $0 \leq t \leq t_{\max}$. Furthermore, we necessarily have to consider a discrete grid within this continuous space-time interval. We choose a simple $(N+1)$ by $(M+1)$ rectangular grid, with step sizes $\Delta t = t_{\max}/N$ and $\Delta x = 2x_{\max}/M$. Points on this grid can be denoted by $(n,j)$ with $0 \leq n \leq N$ and $-M/2 \leq j \leq M/2$. These coincide with the continuous space-time coordinates at $t_n = n\Delta t$ and $x_j = j\Delta x$. To compute derivatives on the discrete grid, we employ a method of finite differences:

$$
\left.\frac{\partial \varphi}{\partial t}\right|_{(t,x)=(t_n,x_j)} \approx \frac{\varphi_j^n - \varphi_j^{n-1}}{\Delta t}, \qquad \left.\frac{\partial^2 \varphi}{\partial t^2}\right|_{(t,x)=(t_n,x_j)} \approx \frac{\varphi_j^{n+1} - 2\varphi_j^n + \varphi_j^{n-1}}{\Delta t^2},
$$

$$
\left.\frac{\partial \varphi}{\partial x}\right|_{(t,x)=(t_n,x_j)} \approx \frac{\varphi_j^n - \varphi_{j-1}^n}{\Delta x}, \qquad \left.\frac{\partial^2 \varphi}{\partial x^2}\right|_{(t,x)=(t_n,x_j)} \approx \frac{\varphi_{j+1}^n - 2\varphi_j^n + \varphi_{j-1}^n}{\Delta x^2}. \tag{A.19}
$$

The equation of motion for field values $\varphi_j^n$ on the discrete grid can now be written as:

$$
\varphi_j^{n+1} = 2\varphi_j^n - \varphi_j^{n-1} + \frac{\Delta t^2}{\Delta x^2}(\varphi_{j+1}^n - 2\varphi_j^n + \varphi_{j-1}^n) - \Delta t^2 \left.\frac{dU}{d\varphi}\right|_{\varphi=\varphi_j^n}. \tag{A.20}
$$

The initial conditions required to solve the complete dynamics then consist of the field configurations $\varphi_j^0$ and $\varphi_j^1$ at the initial two time slices. This can be thought of as specifying a position (amplitude) and velocity (rate of change) for all classical degrees of freedom (the field values) in the classical field theory. In general, the numerical simulation is expected to be stable on the chosen rectangular grid as long as the temporal step size is smaller than the spatial one [82].

A peculiarity of the finite differences used to compute spatial derivatives, is that they cannot be reliably computed at the edge of space because no information is available about the field

values at $|j| = M/2 + 1$. Physically, this means that any propagating wave in the field that reaches the edges of space will be reflected by the boundaries. To avoid seeing these unphysical reflections in the final simulated dynamics, the output of the code should only contain field values at positions $|j| < M/2 - N$. In this way, even the mistake made in the spatial derivative at time step $n = 0$, or equivalently the wave that is reflected at the edge of space at $t = 0$, has no time to propagate into the observed spatial interval.

To ensure that the numerical code functions correctly and that the inherent numerical error of the calculation does not get out of hand, several consistency checks can be considered. First of all, static solutions like $\varphi = \pm 1$ or the kink solution should not change in time. Secondly, the total energy should be conserved in time, except for the energy flowing out of the window of observation defined by $|j| < M/2 - N$. This can be checked by computing the change in total energy, which should be close to zero:

$$\Delta E_{\text{tot}}(n') = E(n = n') - E(n = 0) - \sum_{n=0}^{n'} \left[ \frac{\partial \varphi}{\partial t} \frac{\partial \varphi}{\partial x} \right]_{j=-M/2+N}^{M/2-N}$$

$$\text{with} \quad E(n) = \sum_{j=-M/2+N}^{M/2-N} \left( \frac{1}{2} \left( \frac{\partial \varphi}{\partial t} \right)^2 + \frac{1}{2} \left( \frac{\partial \varphi}{\partial x} \right)^2 + \frac{1}{4} \left( 1 - \varphi^2 \right)^2 \right). \tag{A.21}$$

In these expressions, the partial derivatives should be interpreted in terms of the finite differences defined in Eq. (A.19). The conservation of energy ($\Delta E_{\text{tot}} \simeq 0$) can be checked at every time step in the simulation, and provides a quantitative measure for the accuracy of the simulation.

The final consistency check is to ensure that the field dynamics obtained in the numerical calculation does not change when decreasing the step sizes $\Delta t$ and $\Delta x$. The actual values of the step sizes can then be chosen empirically such that they ensure a desirable balance between having a manageable run time for the numerical code, and an adequate level of accuracy in the results. To quantify the latter, we should use both the independence of the result from the value of step sizes and the conservation of total energy. To simulate two-kink collisions in $\varphi^4$-theory, it turns out that $\Delta t = 0.009$ and $\Delta x = 0.01$ are reasonable values [45]. To simulate kink-impurity interactions the values $\Delta t = 0.01$ and $\Delta x = 0.02$ seem more appropriate.

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
