# Peer review of "An introduction to kinks in $\varphi^4$-theory"

_SciPost Physics Lecture Notes, doi:SciPost Phys. Lect. Notes 23 (2021)_

## Round 1 · Referee Report · Anonymous (Referee 1) · 2020-11-8

Strengths
pedagogical, summary of $\phi^4$ kinks. They have managed to do so
without neglecting mathematical aspects, including some that were new
to me (the elliptic sine solution). The standard of english is excellent.
Report
pedagogical, summary of $\phi^4$ kinks. They have managed to do so
without neglecting mathematical aspects, including some that were new
to me (the elliptic sine solution). The standard of english is excellent.
Requested changes
I would change the title of Section 3 to "Kink-antikink collisions".
On page 12, replace "loose" by "lose".
Although it is difficult to cite all literature, I do feel that
the tutorial would be enhanced by mentioning the following:
@article{kevrekidis2019dynamical,
title={A Dynamical Perspective on the $\varphi^4$ model},
author={Kevrekidis, Panayotis G and Cuevas-Maraver, Jes{\'u}s},
journal={Past, present and future. Nonlinear Systems and Complexity},
volume={26},
year={2019},
publisher={Springer}
}
@article{rice1983physical,
title={Physical dynamics of solitons},
author={Rice, MJ},
journal={Physical Review B},
volume={28},
number={6},
pages={3587},
year={1983},
publisher={APS}
}
@article{bishop1980solitons,
title={Solitons in condensed matter: a paradigm},
author={Bishop, AR and Krumhansl, JA and Trullinger, SE},
journal={Physica D: Nonlinear Phenomena},
volume={1},
number={1},
pages={1--44},
year={1980},
publisher={Elsevier}
}
@article{goodman2005kink,
title={Kink-Antikink Collisions in the $\phi^4$ Equation: The n-Bounce Resonance and the Separatrix Map},
author={Goodman, Roy H and Haberman, Richard},
journal={SIAM Journal on Applied Dynamical Systems},
volume={4},
number={4},
pages={1195--1228},
year={2005},
publisher={SIAM}
}

Anonymous on 2020-11-10 [id 1044]
This is an interesting and useful review article which is very carefully drafted.
This review deals solitons arising in $\phi^4$ which is not integrable and hence interactions between the kinks lead to interesting behaviour in comparison to the (integrable) sine-Gordon model. Major focus of the review is on the dynamics and interactions of kinks in the $\phi^4$ theory. The basic theory of solitons and and kink collisions are nicely discussed in chapters 2 and 3. Collective coordinate approximation, gluing static solutions and kink-impurity interactions are discussed in subsequent sections. Overall, the review is self-sufficient and mathematical steps and arguments are quite transparent. The discussion in the main text is supplemented by three appendices. Indeed, authors have discussed a wide range of using numerical calculations and effective models.
I have just two suggestions:
2.Secondly, I find this article a bit dry. This is not a criticism but, I do believe, review would look far more enriched if examples of physical situations are added wherever possible. For example, as given in Chaikin Lubensky chapter 9. The article would then attract a wider audience.
With these minor comments, I would accept this review for publication.

---

## Round 1 · Referee Report · Anonymous (Referee 2) · 2020-11-10

Strengths
Weaknesses
Report
This is an interesting and useful review article which is very carefully drafted.
This review deals solitons arising in $\phi^4$ which is not integrable and hence interactions between the kinks lead to interesting behavior in comparison to the (integrable) sine-Gordon model. Major focus of the review is on the dynamics and interactions of kinks in the $\phi^4$ theory. The basic theory of solitons and and kink collisions are nicely discussed in chapters 2 and 3. Collective coordinate approximation, gluing static solutions and kink-impurity interactions are discussed in subsequent sections. Overall, the review is self-sufficient and mathematical steps and arguments are quite transparent. The discussion in the main text is supplemented by three appendices. Indeed, authors have discussed a wide range of using numerical calculations and effective models.
I have just two suggestions:
- It is very heartening to that authors have provides some relevant exercise in every section? What about providing some hints and answers at the end
2.Secondly, I find this article a bit dry. This is not a criticism but, I do believe, review would look far more enriched if examples of physical situations are added wherever possible. For example, as given in Chaikin Lubensky chapter 9. The article would then attract a wider audience.
With these minor comments, I would accept this review for publication.
Requested changes
Already in report

---

## Round 2 · Referee Report · Anonymous (Referee 1) · 2021-1-5

Report

I am happy with the small changes made.

---

## Round 2 · Referee Report · Amit Dutta (Referee 3) · 2021-1-13

Report

I have gone through the revised version. The authors have appropriately
taken care of my comments. I therefore, recommend its publication
in the present form.

---

## Round 2 · Author Response

Dear editor,

we are thankful to the referees for their positive reports and supportive comments.
Please find below a list of changes we made to the manuscript, addressing all of the minor issues raised by the referees.

Best regards,
Mariya Lizunova and Jasper van Wezel

---

## Round 2 · List of Changes

We would like to thank both referees for their positive comments and useful suggestions. In response to their remarks, we made the following changes to the manuscript:

First referee: 1. We thank the referee for pointing out the typo, and changed "loose" into "lose", as suggested. 2. We thank the referee for proposing the change in section name, and adopt their suggestion. 3. We have added all recommended citations in relevant places in the text.

Second referee: 1. We appreciate the suggestion of the referee that hints and solutions to exercises may be useful. However, several exercises already have the answers outlined in the main text, and others are of a more practical nature, asking students to produce their own numerical codes. To stimulate actual effort from students in working through derivations and writing code, we prefer not to provide detailed answers to most exercises. We do however appreciate the usefulness of some further help along the way, and we therefore added hints to Exercises 3.1, 4.1, and 6.1.

  1. Although we agree that discussions of physical applications can be useful and interesting, the choice not to include them in the present lecture notes was intentional. The aim of the lecture notes is to provide the mathematical and computational basis needed for understanding kink dynamics and its simulations in general, without the bias of having an application in mind. The vast scala of applications in which these dynamics feature can be easily be explored by interested students, starting from the citations that we provide in the Introduction.

---

## Editorial Decision

published